# GCACL-Rec: A study on conversational recommendation via global context-aware and multi-view contrastive adversarial joint learning

Xianghui Li, Xiaowen Liu, Xinhuan Chen, Ming Ma*

School of Computer Science and Technology, Beihua University, JiLin, China

* ma@beihua.edu.cn

**Data availability statement:** The data used in this study are publicly available on the GitHub platform. The datasets include: Diginetica:

## Abstract

Session-based recommendation (SBR) aims to provide personalized recommendations based on anonymous user click sequences. Although existing methods have achieved notable progress, most focus solely on user preferences within a single session, overlooking item transitions across sessions, which limits their ability to model complex behavior patterns. To address this, we propose GCACL-Rec, a model that enhances dynamic modeling by incorporating global item transition information. It constructs a multi-scale graph structure using Multi-scale graph neural networks (MSGNN) and introduces a relative multi-head attention mechanism (RMA) to enhance cross-session dependency modeling. In addition, a multi-view contrastive-adversarial joint learning strategy (MPACL) is adopted to distinguish better relevant from irrelevant information and extract user intent more effectively. During prediction, we use a hybrid structure that combines a neural decision forest (NDF) with the softmax function to enable fine-grained decision optimization and improve feature discrimination and accuracy. Experiments on the Diginetica, Tmall and RetailRocket benchmark datasets show that GCACL-Rec outperforms existing methods, demonstrating clear advantages in cross-session recommendation tasks.

## Introduction

In the Internet era, information overload has become increasingly prominent. Recommender systems address this challenge by analyzing user behavior to deliver personalized suggestions. Traditional methods, such as collaborative filtering [1–3], depend on users' long-term historical data. However, in domains such as e-commerce and streaming media, this data is often limited or unavailable. In contrast, recent user behaviors and trending clicks more accurately reflect current interests but are typically overlooked by conventional approaches. To address this, session-based recommendation (SBR) has emerged, which predicts the following item of interest for anonymous users by analyzing their click sequences. This approach has gained significant attention in recent years [4,5].

https://github.com/RecoHut-Datasets/diginetica.
Tmall: https://github.com/Ryan-FanZhang/
Tmall-Datasets. Retailrocket:
https://github.com/guocheng2025/
Sequential-Recommendation-Datasets. The
datasets comprise various image sets used for
model training, validation, and evaluation,
covering diverse art styles, scenes, and
emotionally evocative images. All relevant data
are available without restrictions, ensuring the
reproducibility of the study.

**Funding:** This research was supported by the
2024 Graduate Student Innovation Program of
Beihua University (Project No. 2024058). The
funding provided support for data collection,
basic research activities, and partial
dissemination of academic findings. The
funders had no role in study design, data
collection and analysis, decision to publish, or
preparation of the manuscript. There was no
additional external funding received for this
study.

**Competing interests:** The authors have
declared that no competing interests exist.

Due to its substantial practical value, SBR has attracted growing academic interest, resulting in various methodological developments. Some early studies (e.g., Shani et al., 2002; Rendle et al., 2010) applied Markov chain models [6] to infer user behavior based on recent interactions. However, these models rely on strong independence assumptions, resulting in oversimplified behavior patterns that fail to accurately reflect the complex dynamics of user intent, often limiting prediction accuracy.

In recent years, deep learning-based session-based recommendation (SBR) methods have made significant progress [7–9], primarily focusing on extracting item interaction patterns within sessions. Current approaches follow two main directions. Sequential models, typically based on RNNs [10,11], capture the dynamic evolution of user behavior and often employ attention mechanisms [12] and memory units to model long-term interests. Structural models, on the other hand, utilize GNNs [13–17] to build item graphs, with architectures such as GGNN [15] and GAT [18] uncovering latent item relationships.SR-GNN [15] introduces a gating mechanism to guide information flow, effectively modeling item transitions. However, its focus on the final clicked item limits its ability to capture the full sequence structure. FGNN [16] addresses this issue by modeling the entire item sequence, resulting in improved performance in both accuracy and user satisfaction. Despite these advances, current models face two key limitations: (1) they rely on single-session data and ignore cross-session relationships, and (2) they model sessions as directed subgraphs, reducing item transitions to pairwise interactions and missing higher-order patterns. These issues hinder the full understanding of user behavior, particularly in complex scenarios involving cross-session activity or multi-item combinations. Therefore, developing models that integrate both intra-session dynamics and inter-session relations has become an urgent research need [19,20].

To address the above challenges, we propose GCACL-Rec (Global Context-Aware Contrastive Learning for Recommendation), a novel session-based recommendation model that integrates multiple components for cross-session modeling, contrastive learning, and decision fusion. At its core, GCACL-Rec includes a global-level Multi-Scale Graph Neural Network (MSGNN) to capture complex item transitions across sessions and a local-level Position-aware Graph Neural Network (P-GNN) to model sequential dependencies within individual sessions. MSGNN incorporates external hypernodes that aggregate semantically related sessions, enabling cross-session message passing through node–hypernode interactions. To further enhance this process, a relative multi-head attention mechanism is applied to capture position-sensitive dependencies across nodes. For local modeling, P-GNN leverages positional encoding to effectively capture sequential dependencies within sessions. Additionally, we introduce MPACL, a Multi-Perspective Adversarial Contrastive Learning framework that constructs diverse session views and uses adversarial training to maximize inter-view consistency, improving feature robustness and discrimination. In the prediction stage, inspired by SR-PredAO [21], we design a hybrid module that combines Neural Decision Forests (NDF) with a softmax function. NDF captures complex nonlinear transitions, while softmax maintains linear interpretability, jointly enhancing the model's ability to learn diverse item transition patterns.

In summary, the main contributions of our work are as follows:

- We propose a novel multi-scale graph construction and MSGNN model that effectively captures complex and high-order item transitions. This model is enhanced by a relative multi-head attention mechanism, which improves node information flow.
- We design MPACL, a multi-perspective adversarial contrastive learning framework that builds multiple session views and uses adversarial training to ensure cross-view consistency.

- We introduce a hybrid prediction module combining Neural Decision Forests (NDF) and softmax, enabling the model to capture diverse and nonlinear item transitions.
- We validate GCACL-Rec on three benchmark datasets, demonstrating that it consistently outperforms existing state-of-the-art methods in the session-based recommendation.

## Related work

### Session recommendation based on traditional methods

Session-based recommendation (SBR) systems aim to predict the following item a user will interact with by analyzing short-term behavior sequences. Traditional SBR approaches fall into two main categories: co-occurrence-based methods [22] and Markov chain-based methods [6,23]. Co-occurrence methods leverage item-to-item similarity to recommend frequently co-appearing items but struggle to model sequential dependencies. In contrast, Markov chain models capture short-term preferences from recent clicks and effectively reflect immediate user intent. However, they often overemphasize the latest interactions and overlook long-term user preferences. These limitations have motivated the development of more advanced methods that provide a deeper understanding of user behavior and enhance recommendation performance.

### Deep learning based session recommendation

Deep learning has recently driven significant advances in session-based recommendation (SBR). As a pioneering work, Hidasi et al. [10] first applied a recurrent neural network (RNN) to sequential recommendation, introducing the GRU4Rec model, which laid the foundation for future studies. Building on this, Li et al. [4] enhanced GRU4Rec with an attention mechanism to better capture representative item transitions. Parks et al. [5] further advanced the field by proposing STAMP, replacing RNN encoders with attention to jointly model users' long-term preferences and short-term behaviors. In terms of model architecture innovation, Wang et al. [24] proposed SASRec, which employs self-attention to extract deep transition patterns in item sequences. Despite these developments, limitations remain. Most RNN— and CNN-based models emphasize direct transitions between adjacent items, often neglecting complex, latent relationships among non-adjacent items, such as cross-session dependencies or multi-hop neighbors [25,26]. This restricts their ability to capture user behavior patterns fully. Such architectural limitations become especially problematic in cross-session scenarios, where models may fail to reflect users' valid preferences. Overcoming these challenges is, therefore, a key focus for future research in SBR.

### Session recommendation based on graph neural networks

In recent years, graph neural networks (GNNs) have shown strong performance in capturing complex node relationships [27–29]. In the session-based recommendation (SBR) field, Wu et al. [15] pioneered the use of Graph Neural Networks (GNNs) by introducing SR-GNN, which reformulates the sequential recommendation task as a graph modeling problem. SR-GNN leverages a gated graph neural network (GGNN) to learn item representations and extract high-order information. Inspired by SR-GNN, various GNN-based models have been proposed. GC-SAN [30] combines GGNN with self-attention [31] to capture local item transitions and model long-term interests. NISER [32] utilizes normalization to enhance robustness in the presence of sparse and noisy data. $S^2$-DHCN [33] adopts a dual-channel hypergraph

network to model short-term interests and enhance hypergraph learning. CSRM [34] incorporates contextual factors (e.g., time, location) and attention to improve sequential modeling. SGNN-HN [13] addresses non-adjacent item relations and overfitting using star-shaped GNNs and highway networks. CoSAN [35] enhances session-awareness by utilizing neighborhood session embeddings and multi-head attention. GCE-GNN [36] constructs a global session graph by combining local and global session representations. Beyond recommender systems, graph-based models have also been applied in safety-critical dynamic system prediction tasks. For instance, Peng et al [37] proposed the Graph Time Neural Network (GTN), which integrates graph attention mechanisms with multi-scale time series analysis and auxiliary learning to enhance long-term prediction in train–bridge coupled systems. Similarly, Zhang et al. [38] introduced a self-evolving Graph Isomorphic Network (GIN) for multi-scenario safety assessment of railway bridges, demonstrating robust performance across both known and unseen scenarios. These studies provide new perspectives and inspire our work by showing how graph-based attention and self-evolutionary strategies can effectively address complex sequential dependencies and cross-scenario modeling. In the context of session-based recommendation, Int-GNN [39] represents an attempt to incorporate frequency-based signals to model user intent within sessions, achieving competitive performance. However, consistent with the challenges identified in other domains such as dynamic system prediction and cross-scenario safety assessment, Int-GNN still suffers from key limitations: it focuses only on single-session modeling, overlooks cross-session intent dynamics, and lacks effective strategies for handling data sparsity—one of the most common challenges in SBR. Building on the new perspectives provided by recent graph-based advances in other fields and addressing the specific shortcomings of Int-GNN, our study proposes a more robust and generalizable session-based recommendation model.

## Contrastive learning recommendation

Contrastive Learning (CL) is a self-supervised paradigm that extracts supervisory signals from unlabeled data through pretext tasks. Its core idea is to learn robust representations by maximizing agreement between different augmented views of the same instance (positives) while minimizing similarity with views of other instances (negatives), typically optimized via contrastive loss. CL first achieved breakthroughs in computer vision with models such as SimCLR [40] and MoCo [41], and has since been extended to natural language processing [42] and audio processing [43].

Recently, CL has gained traction in recommender systems for alleviating data sparsity and improving representation learning. LightGCL [44] employs singular value decomposition (SVD) for global alignment, while SimGCL [45] demonstrates that injecting uniform noise into embeddings achieves strong performance at lower cost. Building on these foundations, the principles of CL—view-invariant representation and hardness-aware negative sampling—have been adapted to domain-specific challenges. For example, in knowledge-grounded dialogue (KGD) [46], entity-aware CL improves robustness by constructing positives with irrelevant noise and negatives with relevant noise, while in time series, TimesURL [47] leverages frequency-temporal augmentations and dual negative sampling to learn generalizable representations.

In session-based recommendation (SBR), CL has been applied to handle user anonymity and noisy interactions. SCL [48] simplifies CL by directly optimizing alignment and uniformity without complex augmentations, improving interpretability and efficiency. RecDCL [49] further introduces a dual framework with batch-wise and feature-wise objectives for enhanced robustness. However, most methods still rely on fixed augmentations and overlook adversarial

and multi-perspective alignment. To address this, we propose Multi-Perspective Adversarial Contrastive Learning (MPACL), which builds tailored session views and employs adversarial training to generate challenging negatives, thereby distilling true user intent from noise and improving both robustness and discriminative power of session representations.

## Preliminary

Let $V = \{v_1, v_2, \ldots, v_n\}$ denote the set of all items involved in the sessions, where $n$ is the total number of unique items. A session is represented as $S = \{v_{s,1}, v_{s,2}, \ldots, v_{s,m}\}$, which includes $m$ interactions from an anonymous user, where $v_{s,i} \in V$ indicates the item clicked at the $i$-th step of the $s$-th session. Given a specific session $S$, the goal of session-based recommendation (SBR) is to predict the next item $v_{s,m+1}$. To this end, our model aims to output a probability distribution over all items, denoted as $\hat{y} = \{\hat{y}_1, \hat{y}_2, \ldots, \hat{y}_n\}$, where $\hat{y}_i \in \hat{y}$ represents the predicted score of item $v_i$. These scores are sorted in descending order, and the top-K ranked items are recommended as candidates. The frequency of occurrence of an item is calculated using a count function $Cnt(v_s^i, S_i)$, which measures the number of times the item $v_s^i$ appears in the subsequence $S_i$. The frequency vector of the occurrence of the sequence is defined as: $seqOcc(S) = [Cnt(v_1^s, S_1), \ldots, Cnt(v_n^s, S_n)]$ This vector is used to capture the temporal occurrence patterns of items within a session.

## Method

### Methodological framework overview

The overall framework of GCACL-Rec is illustrated in Fig 1. First, all sessions and the current session are constructed into a global-level graph and a local-level graph, respectively. We also

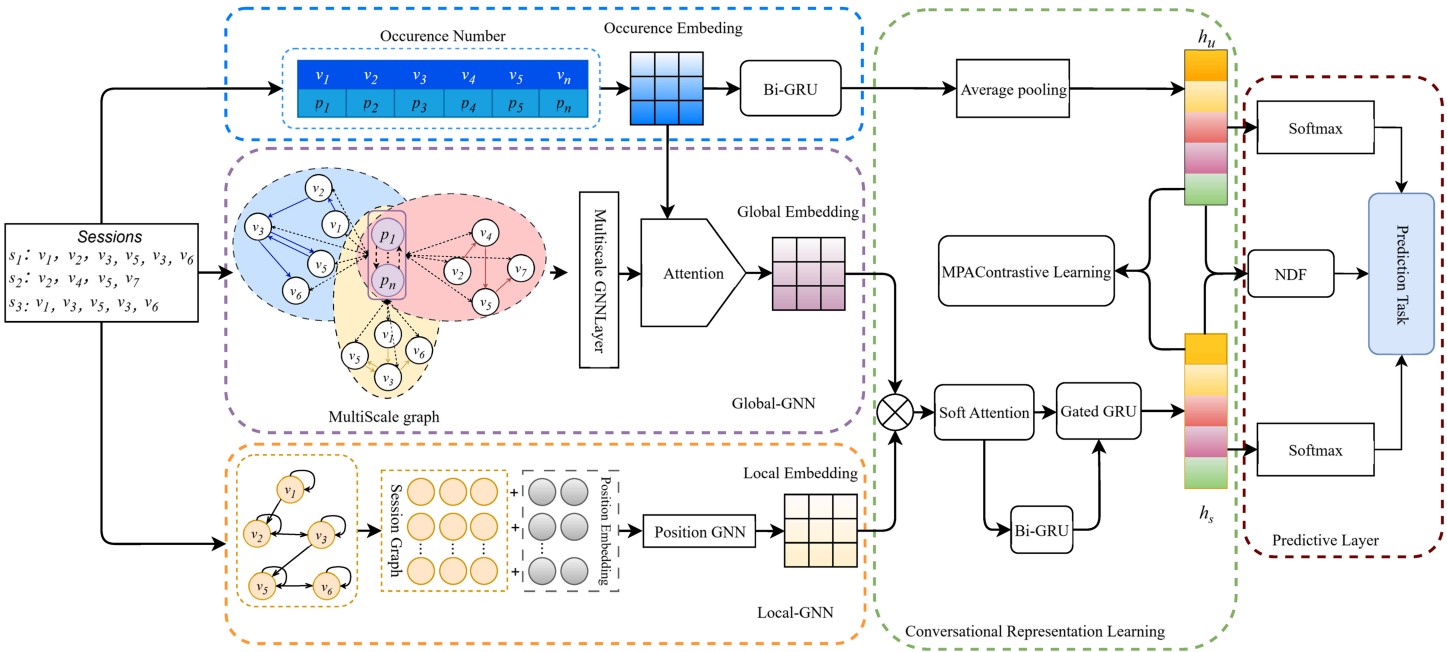

**Fig 1. Overall structure diagram of the GCACL-Rec model.** The diagram provides a high-level overview of the model architecture, aiding understanding before the detailed explanation.

compute item occurrence counts across all sessions to obtain frequency-based item embeddings. The local graph is processed by a Position-aware GNN (P-GNN) to capture high-order local dependencies and generate local item embeddings. The global graph is processed by an MSGNN and, combined with the frequency embeddings, is further refined by a statistics-aware attention mechanism to emphasize high-frequency items, yielding global item embeddings. Next, a soft attention-based fusion module integrates global and local embeddings to learn a pairwise-aware session representation. Meanwhile, the frequency embeddings are passed through a Bidirectional GRU (Bi-GRU) to model both short—and long-term user preferences. Average pooling is followed by deriving a user preference embedding $h_u$, representing the user's interest over the entire item set $V$. The fused item embeddings are also input into another Bi-GRU to capture temporal dependencies further. The original fused embeddings and Bi-GRU outputs are then combined using a gated GRU to retain informative features, resulting in the final item representation $h_s$. Additionally, GCACL-Rec introduces a multi-perspective adversarial contrastive learning strategy, which constructs multiple session views and maximizes their consistency via adversarial training. Finally, a hybrid prediction layer combining Neural Decision Forests (NDF) and Softmax is used to compute the score for each candidate item, predicting the following item the user is likely to click in the given session.

## Local level graph neural network

To capture pairwise relationships between items within a session, we construct a local graph $G_L = (V_L, E_L)$ for each session, where $V_L$ denotes the set of item nodes in the current session and $E_L$ represents the edges based on the item click sequence in session $S$. To better model item transition patterns, a self-loop is added to each item node to capture its self-state. Since the local graph is directed and item transitions are directional, edges are categorised into four types: $r_{in}$ (transition from $v_i$ to $v_j$ only), $r_{out}$ (from $v_j$ to $v_i$), $r_{in-out}$ (bidirectional transitions), and $r_{self}$ (self-loops). This edge typing strategy allows the model to represent item interaction patterns within sessions more accurately.

Building on this, we introduce a positional embedding $pos\_emb_i$ for each node to capture its position within the sequence. Specifically, the positional embeddings are generated as follows:

$$P_i = normalize(W_p \cdot pos\_emb_i) \tag{1}$$

$W_p$ is a trainable embedding matrix that maps each node's position index $pos\_index_i$ to a high-dimensional space. The normalization function $normalise(\cdot)$ stabilizes the embedding values and reduces noise during propagation.

To capture node relationships more effectively, each node $v_i$ updates its features through multiple graph convolution layers, where each layer takes the output of the previous one as input:

$$h_i^l = \sigma\left(\sum_{v_j \in \mathcal{N}_i} A_{ij} P_i^{(l-1)} W_{r_{ij}}^l + b^l\right) \tag{2}$$

Where $A_{ij}$ is the normalized measure of the relation or relative position between nodes $v_i$ and $v_j$, ensuring that the sum of weights over all neighbors of a node equals 1, $r_{ij}$ denotes the connection type between $v_i$ and $v_j$, $P_i^{(l-1)}$ is the feature representation of node $v_i$ at layer $l$–1, and the initial input $P_i^{(0)}$ is the positional embedding of node $v_i$.

After the graph convolution, we use a Gated Recurrent Unit (GRU) to capture time-related patterns. GRUs are suitable for modeling sequences and managing long-term dependencies. The computation is defined as follows:

$$z_t = \sigma\left(W_z \cdot [h_t^l, x_t] + b_z\right)$$
$$r_t = \sigma\left(W_r \cdot [h_t^l, x_t] + b_r\right)$$
$$\tilde{h}_t = \tanh\left(W_h \cdot [r_t \cdot h_{t-1}^l, x_t] + b_h\right)$$
$$h_t = (1 - z_t) \cdot \tilde{h}_t + z_t \cdot h_{t-1}$$

(3)

Where $h_t$ is the hidden state at time $t$, and $x_t$ is the node feature output from the graph convolution.

Finally, after processing through the graph convolution and GRU layers, a fully connected network is used to map and extract node features, producing the final node representations:

$$X_{v_i}^L = W_2 \, \text{ReLU}\left(W_1 h_i + b_1\right) + b_2$$

(4)

Where $h_i$ denotes the feature output by the GRU, and $W$ and $b$ are learnable parameters.

Finally, the project feature representation obtained through local graph modeling is defined as $X_{v_i}^L$, where $X_{v_i}^L$ represents the embedding of the $i$-th project in the local graph.

## Global-level graph neural networks

To address the issue of insufficient item representation in Local-GNN session modeling caused by data sparsity, this study proposes constructing a global graph to integrate transition patterns across all sessions. Specifically, given a session set $S = [S_1, S_2, ..., S_n]$, we construct a global graph $G_g = (V_g, E_g)$, where $V_g$ represents item nodes appearing in all sessions. In this paper, we use the term hypernodes consistently to denote auxiliary nodes that aggregate structurally or semantically similar sessions. Similarly, we use global graph to describe the integrated cross-session structure. To capture high-order item transitions across sessions, a set of hypernodes $P_g = [P_1, P_2, ..., P_m]$ is introduced, each aggregating structurally or semantically similar sessions. For any node $v_i \in V_g$, its neighborhood includes both item nodes with direct transition relationships and connected hypernodes that introduce semantic associations across sessions. Unlike traditional GNNs that model only local structures based on adjacency matrices, we adopt a dynamic routing mechanism between nodes and hypernodes to enable bidirectional information exchange. This design effectively integrates local transition patterns with global semantics, providing a unified and efficient graph structure for modeling cross-session dependencies and representation learning.

**Information propagation between nodes.** In this method, message passing between nodes considers both the current session representation and the weight assignment between a node and its neighbors. Specifically, for node $v_i$ and each of its neighbors $v_j$, an attention score is calculated by combining the feature of $v_j$ with the semantic information of the current session, indicating the importance of $v_j$. The calculation is as follows:

$$b_{ij} = \text{LeakyReLU}\left(W_1 \left(x_{v_i} \odot x_{v_j}\right)\right)$$

(5)

where $\odot$ denotes element-wise multiplication, and $W_1$ is a learnable weight matrix. This design allows the model to better capture the importance of neighbor nodes in the current session.

To ensure the comparability of weights when computing attention for node $v_i$, the attention scores $b_{ij}$ are normalized using *Softmax*:

$$b_{ij} = \frac{\exp\left(\text{LeakyReLU}\left(W_1\left(x_{v_i} \odot x_{v_j}\right)\right)\right)}{\sum_{v_k \in \mathcal{N}^g_{v_i}} \exp\left(\text{LeakyReLU}\left(W_1\left(x_{v_i} \odot x_{v_j}\right)\right)\right)} \tag{6}$$

Where $N^g_{v_i}$ denotes all neighbors connected to node $v_i$. The normalized $b_{ij}$ represents the attention weight of neighbor $v_j$ relative to other neighbors.

After computing and normalizing the attention weights, the updated representation $x^g_{v_{i_s}}$ is obtained by the weighted sum of all neighbor representations of node $v_i$:

$$x^g_{v_{i_s}} = \sum_{v_j \in \mathcal{N}^g_{v_i}} b_{ij} x_{v_i} \tag{7}$$

This process performs weighted aggregation of neighbor features, allowing neighbors more relevant to the current session to contribute more. The resulting representation $x^g_{v_{i_s}}$ combines both the node's information and that of its neighbors, capturing richer semantic features in the graph.

**Information propagation between hypernodes.** To effectively capture high-order graph information, we generate hypernodes through three complementary strategies: mean pooling preserves session-wide statistics, max pooling extracts dominant features, and random sampling (activated when hypernodes exceed three) enhances diversity. These hypernodes then undergo relative multi-head attention, facilitating position-aware multi-round interactions. The initial hypernode representations are defined as $S^{(0)} = [s_1^{(0)}, s_2^{(0)}, ..., s_L^{(0)}]^\top$, where $L$ is the number of hypernodes.

Given the hypernode representations at the $t$-th layer $S^{(t)} \in R^{L \times d}$, we project them into query, key, and value vectors to capture interactions and relative positional information: $Q = S^{(t)} W^Q$, $K = S^{(t)} W^K$, and $V = S^{(t)} W^V$, where $W^Q$, $W^K$, and $W^V$ are learnable parameter matrices. With $H$ attention heads, each head has a dimension of $d_k = d/H$. In the $h$-th head, the relative attention score between hypernodes $i$ and $j$ is computed as follows:

$$A^h_{i,j} = \left(Q^h_i + \mu\right)^\top K^h_j + \left(Q^h_i + \gamma\right)^\top R^h_{i-j} \tag{8}$$

Where $R(i{-}j)$ represents the relative positional information between hypernodes $i$ and $j$, derived from their index difference $i{-}j$, and $\mu, \gamma$ are learnable bias vectors.

Then, $A^h_{i,j}$ is scaled and normalized using *Softmax*, and the weighted sum of the value vectors obtains the output:

$$O^h_i = \sum_{j=1}^{L} \text{Softmax}\left(\frac{A^h_{i,j}}{\sqrt{d_k}}\right) V^h_j \tag{9}$$

With $h$ attention heads, we compute outputs $O^{(1)}, O^{(2)}, ..., O^{(H)}$ and concatenate them along the feature dimension. The concatenated result is then transformed by a learnable linear projection matrix $W^O$ to obtain the updated hypernode representations $P^{(t)}$.

$$p^t = \left[o^1 \| o^2 \| \cdots \| o^h\right] \cdot W^o \tag{10}$$

where $W^O \in \mathbb{R}^{(H \cdot d_k) \times d}$ is a learnable projection matrix. Through multiple iterations, hypernodes exchange information under the guidance of relative positional encoding, leading to representations with enhanced global awareness.

The final hypernode representation $p^{(T)}$ is considered the output that captures both high-order structural information and relative positional dependencies. It retains the local semantic information aggregated within each hypernode while integrating the influence of other hypernodes and their relative positions in the index space, resulting in a globally-aware and hierarchically-structured representation.

**Cross-propagation.** In each propagation round, relying only on node-to-node or hypernode-to-hypernode interactions limits information exchange to shallow structures, making it difficult to integrate global context and local structure effectively. To address this, we design a cross-propagation mechanism that enables nodes to directly receive information from hypernodes, while hypernodes can also capture dynamic feedback from nodes.

Based on the previously obtained node representations $x_{v_i}^g$ and hypernode representations $p^{(t)}$, we introduce a bidirectional propagation path between hypernodes and nodes to enhance global-local fusion. During hypernode updates, we first compute interaction results via relative multi-head attention (RMHA) among hypernodes and then integrate dynamic feedback from nodes. The update is given by:

$$p^{T+1} = \text{RMHA}(p^1, p^2, ..., p^T) + A_{n,s}^\top \cdot x_{v_{i_s}}^g \tag{11}$$

where $A_{n,s}^T \in \mathbb{R}^{M \times N}$ represents the connection matrix between nodes and hypernodes, this design ensures each iteration allows hypernodes to aggregate global semantic information while dynamically sensing connected node features, preventing semantic drift and enhancing representation consistency and timeliness.

In the node update phase, nodes receive messages not only from their neighbors but also from hypernodes to enhance global awareness. The update is computed as:

$$X^T = \sigma \left( W_x \left( A_{n,n} \cdot x_{v_{i_s}}^g + A_{n,s} \cdot p^{T+1} \right) \right) \tag{12}$$

where $A_{n,n} \in \mathbb{R}^{N \times M}$ is the node-to-neighbor adjacency matrix, $W_x$ is a learnable projection matrix, and $\sigma(\cdot)$ is an activation function.

The bidirectional propagation between hypernodes and nodes is repeated for $T = 0, 1, ..., T-1$. The final node representation $X^{(T)}$ combines neighbor and hypernode information, enabling nodes to retain local features while capturing global context for stronger representations.

**Get global embedding.** In earlier sections, we introduced the MSGNN framework and discussed preliminary item representations in session sequences. However, item occurrences in sessions are often imbalanced, with high-frequency items better reflecting user interests. Explicitly incorporating item frequency into graph representation learning can increase the model's attention to these items and improve recommendation performance. Specifically, we define a global frequency embedding matrix $E_O = [e_0^o, e_1^o, ..., e_M^o]$, where each $e_i^o$ is a learnable embedding for items appearing $i$ times, capturing user preference signals. For each item $v_i$ in a session $S = [s_1, s_2, ..., s_L]$, we compute its frequency $\text{Cnt}(v_i, S)$, retrieve the corresponding embedding from $E_O$, and construct the session's frequency embedding matrix:

$$E_O^S = \left[ e_{\text{cnt}(v_s^1, S_1)}^o, e_{\text{cnt}(v_s^2, S_2)}^o, ..., e_{\text{cnt}(v_s^L, S_L)}^o \right]^T \tag{13}$$

We design a statistic-aware attention mechanism to fuse the semantic and statistical features of items. Semantic features capture contextual information, while statistical features (e.g., frequency) reflect the importance of an item within a session. Simple concatenation or averaging may fail to capture their relative contributions; therefore, we introduce an adaptive weighting mechanism to balance the influence of both feature types dynamically.

To achieve this, we apply an attention-based mechanism to compute the contribution of the occurrence embedding $E_O^S$ relative to the item feature $X^T$, estimating the influence of statistical features in the current representation. The computation is as follows:

$$\alpha_i = \frac{\sigma\left(X^\top W + b\right) \cdot e_{\text{cnt}}^o\left(v_s^i, S_i\right)}{\left\| e_{\text{cnt}}^o\left(v_s^i, S_i\right) \right\|_2} \tag{14}$$

where $W \in \mathbb{R}^{d \times d}$ and $b \in \mathbb{R}^d$ are learnable linear mapping parameters, and $\sigma(\cdot)$ is the Sigmoid activation function.

Based on the attention weight $\alpha_i$ computed above, we weight the semantic representation $X^T$ of node $i$ to generate the final item representation vector:

$$X_{v_i}^G = \alpha_i \cdot X^T \tag{15}$$

The resulting $X_{v_i}^G$ is the final global representation of node $v_i$, combining its semantic features and statistical information within the session.

## Conversational representation learning layer

With the help of G-GNN and L-GNN, we obtain both local and global embeddings of items. To effectively fuse local structural and global semantic representations, we introduce a soft attention-based weighted fusion mechanism. This mechanism dynamically adjusts the contribution of each semantic perspective based on contextual features. First, the local and global embeddings are concatenated along the feature dimension:

$$x_{v_i}' = \left[ X_{v_i}^L \parallel X_{v_i}^G \right] \tag{16}$$

where $\parallel$ denotes vector concatenation, $X_{v_i}^L$ is the local item embedding, and $X_{v_i}^G$ is the global embedding. Then, the soft attention mechanism computes the attention weights as follows:

$$\alpha_{LG} = \text{Softmax}\left(q^\top \cdot \sigma\left(W \cdot x_{v_i}' + b\right)\right) \tag{17}$$

Where q is an attention query vector that captures which items are more important in the current context. Finally, we obtain the session representation based on pairwise relations by linearly combining item embeddings with their attention weights.

$$z_i = \sum_{i=1}^{t} \alpha_{LG} \, x_{v_i}' \tag{18}$$

The fused representation $z_i$ captures both global semantic information and short-term dependencies from the local structure.

In sequential modeling, the order of items plays a key role in capturing user preferences. To address this, we introduce a Bidirectional Gated Recurrent Unit (BiGRU) (consistent with Eq (3), just performing forward and reverse calculations) to capture the dynamic evolution

of context from the fused sequence $Z_i = [z_1, z_2, ..., z_s]$. After processing each $z_i$ with BiGRU, we obtain $H_s^{Bi}$, which incorporates both past and future information, enabling the model to capture both short-term transitions and long-term dependencies.

Although the BiGRU output $H_s^{Bi}$ is expressive, its deep transformations may introduce redundancy or noise. To address this, we introduce a Gated Fusion GRU to dynamically balance the contributions of BiGRU outputs and the original inputs, highlighting significant changes while suppressing interference. Specifically, we first concatenate the two to compute the gating weights:

$$\beta_i = \sigma \left( W_g \cdot \left[ z_i \parallel H_s^{B_i} \right] + b_g \right) \tag{19}$$

The gating coefficients $\beta_i \in [0, 1]$ control the distribution of weights of the information between the two sources, and the final fusion output is:

$$h_s = \beta_i \cdot H_s^{B_i} + (1 - \beta_i) \cdot z_i \tag{20}$$

When $g_i$ is large, more emphasis is placed on the dynamic modeling of BiGRU; when it is small, more of the original local–global fused features are preserved.

Ultimately, the final output sequence $h_s = [h_{s,1}, h_{s,2}, ..., h_{s,i}]$ integrates temporal modeling with global structure and local context, serving as a key representation for contrastive learning and score prediction.

To model item frequency and the user interest it reflects, we have constructed the frequency-based embedding sequence $E_S^O$ and input it into a BiGRU to capture its temporal dynamics, yielding $H_u^{Bi} = \text{BiGRU}(E_S^O)$. This design extracts sequential dependencies to identify user attention patterns, enhancing the model's ability to represent repeated clicks and frequent switches. The final frequency-aware user preference $h_u$ is obtained by averaging the BiGRU output.

$$h_u = \frac{1}{L} \sum_{i=1}^{L} H_u^{Bi} \tag{21}$$

## Prediction layer

This section describes how to score candidate items based on the representation of the learned session and the user's preference. After obtaining the final session embedding $h_s$ and global user preference $h_u$, we design a multi-branch prediction layer to combine user intent and item features for more accurate recommendations. The layer comprises two main components: a traditional softmax-based branch and a high-capacity neural decision forest (NDF) [21] branch, both of which contribute to the final prediction. The scoring method of the traditional branch is as follows:

$$\hat{y}_{v_i} = \text{Softmax} \left( E_S^O (\tilde{h}_u)^\top + E_S^I (\tilde{h}_s)^\top \right) \tag{22}$$

Where $\tilde{h}_u$ and $\tilde{h}_s$ are the L2-normalized user preference and session representations, $E_S^O$ denotes the frequency distribution vector of the item, which captures user tendencies, and $E_S^I$ denotes the embedding of the initial feature of the item based on the session.

However, Most session recommenders use an encoder–predictor setup: the encoder turns a session into a vector, and a predictor scores items. A common linear dot-product predictor has low capacity and struggles with noisy or random clicks, bottlenecking even

strong encoders. Recent work demonstrates that incorporating a high-capacity, tree-inspired module—alongside denoising the session vector and pruning to manage capacity—consistently improves accuracy. Following this, we utilize a global, context-aware encoder and replace the linear head with a Neural Decision Forest (NDF) that incorporates denoising, allowing the final decision boundary to better separate true next-item intent from noise. The concatenated feature from $h_u$ and $h_s$ is first processed with James-Stein shrinkage to reduce variance from small sample sizes:

$$\hat{\mu}_{ij}^{(JS)} = \left(1 - \frac{m-2}{\|\delta_j\|^2}\right) z_{ij} \tag{23}$$

where $z_{ij}$ is the $j$-th feature value, $m$ is the batch size, and $\|\delta_j\|$ is the L2 norm of the $j$-th feature across the batch.

The adjusted vector $z^{(JS)}$ is fed into the Neural Decision Tree (NDT) to compute the probability $p_k^{(\text{leaf})}$ of the sample reaching each leaf node, which is then weighted by the corresponding leaf distribution $\pi_k$ to produce the final output:

$$\hat{p} = \sum_{k=1}^{2^d} p_k^{(\text{leaf})} \cdot \text{Softmax}(\pi_k) \tag{24}$$

Multiple NDTs are ensembled to form the NDF, and their outputs are averaged:

$$\hat{y}_{\text{NDF}} = \frac{1}{T} \sum_{i=1}^{T} \hat{p}_i \tag{25}$$

Where $T$ is the number of trees, and $\hat{p}_i$ is the output of the $i$-th tree. Finally, we fuse the softmax and NDF branches:

$$\hat{y} = q \cdot \hat{y}_{v_i} + (1-q) \cdot \hat{y}_{\text{NDF}} \tag{26}$$

This hybrid design combines user preferences, item features, and high-capacity nonlinear modeling, improving performance and robustness in complex scenarios.

## Model optimization

In this section, we present the optimization process of our model. Specifically, we define the learning objective as the cross-entropy loss function, which has been extensively used in recommendation systems:

$$L_r = -\sum_{i=1}^{n} \left[ y \log(\hat{y}) + (1-y) \log(1-\hat{y}) \right] \tag{27}$$

Where y denotes the one-hot encoding vector of the ground truth item.

## Multi-perspective comparison adversarial joint learning

We propose a joint optimization framework combining multi-view contrastive learning and adversarial training to enhance session representation learning. The model captures user behavior from two complementary perspectives: global session embedding $h_s$ and user intent representation $h_u$, addressing data sparsity and behavioral diversity.

To improve consistency between these views, we introduce a contrastive learning task. Two positive sample views $(s^{(1)}, v^{(1)})$ and $(s^{(2)}, v^{(2)})$ are generated using augmentation strategies such as dropout, noise injection, and sequence shuffling. Normalized cosine similarity is used to measure alignment, and a dynamic temperature scaling mechanism is applied to stabilize gradients. For negative sampling, we employ hard negative mining by computing the cosine similarity between the anchor and all candidates in the batch, selecting the most similar ones as hard negatives to enhance the discriminative power of the contrastive loss.

Furthermore, to further improve representation discrimination, we introduce a Cross-view Shuffling strategy, which randomly shuffles the order of augmented views to create cross-sample negatives. This enhances sample diversity and reduces overfitting to specific patterns. We also retain Random Negative Sampling to ensure training stability. The final contrastive loss is defined as:

$$L_{\text{con}} = -\log\left(\sigma\left(\text{sim}(s^{(1)}, v^{(1)})\right)\right) - \log\left(1 - \sigma\left(\text{sim}(s^{(2)}, v^-)\right)\right) \tag{28}$$

After contrastive learning, we introduce adversarial training to enhance model expressiveness and generalization. A GRU-based generator produces fake sequence embeddings from randomly initialized inputs, refined by a projection head to ensure semantic consistency. A GRU-based discriminator distinguishes between real embeddings $h$ and generated embeddings $\hat{h}$. To improve training stability, we adopt Least Squares GAN (LSGAN) [50] as the loss function:

$$L_D = \frac{1}{2}\left[\left(D(h) - 1\right)^2 + \left(D(\hat{h})\right)^2\right], \quad L_G = \left(D(\hat{h}) - 1\right)^2 \tag{29}$$

In this setup, the generator and discriminator form an adversarial game in the semantic space. Unlike traditional GANs, LSGAN uses a least squares loss to mitigate vanishing gradients and accelerate convergence. The generated embeddings also participate in contrastive learning to improve semantic alignment. The final objective jointly optimizes recommendation, contrastive learning, and adversarial training:

$$L_{\text{total}} = L_r + \mu\left((1 - \beta) \cdot L_{\text{con}} + \beta \cdot (L_D + L_G)\right) \tag{30}$$

where $L_r$ is the main recommendation loss, and $\mu$, $\beta$ control the task balance.

This adversarial setup, combined with multi-view contrastive learning, enhances the model's discriminative power, generative capability, and generalization. The process of GCACL-Rec is detailed in Algorithm 1.

## Results

In this section, we systematically validate the proposed model through a large number of experiments and provide an in-depth analysis of the following core issues:

- RQ1: Is GCACL-Rec better than existing methods?
- RQ2: Are these components necessary in GCACL-Rec?
- RQ3: Is the proposed MSGNN valid?
- RQ4: What is the effect of different numbers of hypernodes?
- RQ5: What is the effect of model depth?
- RQ6: What are the effects of different hyperparameters in GCACL-Rec?

**Algorithm 1.** Training Process of GCACL-Rec

**Input:** Sessions $S$, candidate items $V$
**Output:** Top-$k$ recommended items
1 Transform sessions into local/global views;
2 Construct local graphs $G_L$ and global graph $G_G$ with hypernodes;
3 **for** *epoch in range(Epoches)* **do**
4 **for** *batch in DataLoader* **do**
5 **foreach** *session S in batch* **do**
6 **Graph Modeling:**;
7 Local P -GNN embedding $X_{v_i}^L \leftarrow$ (Eq. (1)-(4));
8 Global MSGNN embedding $X^T \leftarrow$ (Eq. (5)-(12));
9 Statistic-aware refinement $X_{v_i}^G \leftarrow$ (Eq. (13)-(15));
10 **Representation Learning:**;
11 Fusion & sequential encoding $h_s \leftarrow$ (Eq. (16)-(20));
12 Frequency-based preference $h_u \leftarrow$ (Eq. (21));
13 **Multi-Perspective Contrastive & Adversarial Learning:**;
14 Contrastive on $(h_s, h_u) \Rightarrow \mathcal{L}_{con} \leftarrow$ (Eq. (28));
15 Adversarial training $\Rightarrow \mathcal{L}_D, \mathcal{L}_G \leftarrow$ (Eq. (29));
16 **Prediction:**;
17 Hybrid scoring $\hat{y} \leftarrow$ (Eq. (22)-(26));
18 **end for**;
19 **end for**;
20 **Loss:** $\mathcal{L}_{\text{total}} = \mathcal{L}_r + \mu((1-\beta)\mathcal{L}_{con} + \beta(\mathcal{L}_D + \mathcal{L}_G))$ ;
21 Update parameters by backpropagation;
22 **end for**;

## Datasets and preprocessing

To comprehensively evaluate the effectiveness of the proposed method, we conducted experiments on three real-world datasets: Diginetica, Tmall, and Retailrocket. Diginetica is from the CIKM Cup 2016 and contains five months of transaction logs from an e-commerce platform. Tmall is derived from the IJCAI-15 competition and consists of anonymized shopping logs. The RetailRocket dataset, released by an e-commerce company on Kaggle, includes six months of user browsing activities.

For data preprocessing, we follow a unified strategy used in prior work [15]. Sessions with only one item and items appearing fewer than five times are removed. For Tmall, sessions longer than 40 are also filtered out. Training samples are generated using sequence splitting: for a session $S = [s_1, s_2, ..., s_n]$, we construct subsequences like $([s_1], s_2), ([s_1, s_2], s_3), ..., ([s_1, ..., s_{n-1}], s_n)$ for both training and testing. Statistics of the preprocessed datasets are shown in Table 1.

**Table 1**. Dataset statistics on Diginetica, Tmall, and RetailRocket.

| Statistics | Diginetica | Tmall | RetailRocket |
|---|---|---|---|
| #clicks | 982,961 | 818,479 | 1,085,217 |
| # training sessions | 719,470 | 351,268 | 433,643 |
| # test sessions | 60,858 | 25,898 | 15,132 |
| # items | 40,728 | 30968 | 36,968 |
| average lengths | 5.12 | 6.69 | 5.43 |

**Note**: This table summarizes the key statistics of the three benchmark datasets used in the experiments, including the number of clicks, training and test sessions, item counts, and average session length.

### Evaluation indicators

Following the previous work [15,33], we use two relevance-based evaluation metrics: Precision@N (P@N) and Mean Reciprocal Rank@N (MRR@N) to assess model performance. P@N measures the proportion of correctly recommended items in the top-N list, while MRR@N calculates the average reciprocal rank of the correct item within the top-N positions. In this study, we set N = 10 and N = 20 for consistent evaluation across all compared methods.

### Baseline models

To evaluate the model's performance, we comprehensively evaluate the performance of GCACL-Rec with 14 baselines from 5 different types:

#### Methods based on frequency and matrix decomposition

- POP: It employs item frequency statistics to recommend the top-N most popular items from the training set.
- FPMC [6]: It employs matrix factorization combined with a Markov chain to capture short-term item transitions.

#### RNN modeling methods

- GRU4Rec [10]: It is an RNN-based model that uses a Gated Recurrent Unit (GRU) to model user sequences.
- NARM [4]: It employs an RNN encoder to capture sequential context, followed by an attention mechanism to focus on dominant user intent.

#### Attention mechanism methods

- STAMP [5]: It employs attention layers to replace all RNN encoders in previous work by fully relying on the self-attention of the last item in the current session to capture the user's short-term interest.

#### Transformer-based methods

- SASRec [24]: It employs stacked self-attention layers with positional encoding to capture sequential dependencies, followed by unidirectional modeling to predict the next item in the sequence.
- BERT4Rec [51]: It employs bidirectional Transformer encoders with a masked item prediction objective, followed by contextualized embeddings to capture both left and right dependencies for sequential recommendation.

#### Graph neural network methods

- SR-GNN [15]: It employs a gated GNN layer to obtain item embeddings, followed by a self-attention of the last item as STAMP [5] does to compute the session-level embeddings for session-based recommendation.
- NISER [32]: It employs GNN-based item modeling, followed by L2 regularization, to mitigate the effects of long-tail distribution.
- LESSR [52]: It employs a GRU-enhanced graph structure to learn item representations, followed by session-level aggregation.

- GCE-GNN [36]: It employs global and session graphs jointly to learn item embeddings, followed by a fusion layer to enhance the cross-session context.
- S$^2$-DHCN [33]: It employs a hypergraph and line graph to model high-order relations, followed by contrastive learning to optimize representations.
- COTREC [53]: It employs dual graph encoders trained with self-supervised signals, followed by fusion to model intra- and inter-session structures.
- Int-GNN [39]: It employs frequency, re-interaction gaps, and user preferences to guide intent modeling, followed by GNN-based feature extraction.

These baselines span from simple heuristics to advanced GNN and intent-aware methods, enabling a comprehensive evaluation of our model's effectiveness in session-based recommendation.

## Parameter setup

To ensure a fair comparison, we adopt the following experimental settings: 10% of the training set is randomly selected as the validation set, and all models share the same hyperparameter configurations. Specifically, the latent vector dimension is set to d = 256, the batch size is 512, and cross-entropy is used as the loss function. The number of nodes in the neural decision forest is set to 128, and the depth of the tree is set to 4. All model parameters are initialized using a Gaussian distribution $\mathcal{N}(0, 0.1)$. The Adam optimizer [20] is applied with a learning rate of 0.0015, a decay rate of 0.5 every five epochs, and L2 regularization set to $10^{-6}$. Additionally, the maximum item occurrence count $M$ is set to 300, the maximum session length $N$ is 100, and the temperature parameter $\mu$ is 12.5. The number of hypernodes $\eta$ is set to 4 and the contrastive learning weight is set to 0.1. All baseline methods are run using the best-performing settings reported in their original papers, and we report their best results.

## RQ1-performance evaluation

As shown in Table 2, we compare the performance of GCACL-Rec with traditional methods across three widely used benchmark datasets(with the best results highlighted in bold). The results show that GCACL-Rec consistently outperforms all baselines on every dataset.

We first compare two traditional recommendation methods: POP and FPMC. POP recommends the top-N most frequent items based on popularity and shows the weakest performance. FPMC combines first-order Markov chains with matrix factorization to utilize session context for recommendation. While it performs better than POP, it still fails to capture sequential patterns within sessions, leading to suboptimal results.

Compared with neural network-based session recommendation methods, traditional models show apparent limitations in modeling sequential dependencies. GRU4Rec was the first to apply gated recurrent units (GRU) to user behavior sequences, initiating the use of deep learning in session-based recommendation. Later methods, such as NARM and STAMP, introduced attention mechanisms to focus on key behaviors, significantly improving performance. However, they mainly capture explicit interactions while ignoring the implicit relationships between items. Subsequent Transformer-based approaches, such as SASRec and BERT4Rec, further advanced sequential modeling by leveraging self-attention to capture long-range dependencies. While SASRec employs an unidirectional encoder to predict the next item, BERT4Rec adopts bidirectional context through a masked prediction objective. However, both methods still face challenges in modeling higher-order item relations and session-level intent.

**Table 2.** The performance of various models was compared using P@K and MMR@K metrics.

| Datasets | Diginetica | | | | Tmall | | | | RetailRocket | | | |
|---|---|---|---|---|---|---|---|---|---|---|---|---|
| Metrics | P@20 | MRR@20 | P@10 | MRR@10 | P@20 | MRR@20 | P@10 | MRR@10 | P@20 | MRR@20 | P@10 | MRR@10 |
| POP | 1.18 | 0.28 | 0.76 | 0.26 | 2.00 | 0.90 | 1.67 | 0.88 | 1.12 | 0.30 | 0.61 | 0.27 |
| FPMC | 22.14 | 6.66 | 15.43 | 6.20 | 16.06 | 7.32 | 13.10 | 7.12 | 32.37 | 13.82 | 25.99 | 13.38 |
| GRU4Rec | 29.45 | 8.33 | 17.93 | 7.33 | 10.93 | 5.89 | 9.47 | 5.78 | 44.01 | 23.67 | 38.35 | 23.27 |
| NARM | 49.70 | 16.17 | 35.44 | 15.13 | 23.30 | 10.70 | 19.17 | 10.42 | 50.22 | 24.59 | 42.07 | 24.88 |
| STAMP | 45.64 | 14.32 | 33.98 | 14.26 | 26.47 | 13.36 | 22.63 | 13.12 | 50.96 | 25.17 | 42.95 | 24.61 |
| SASRec | 48.78 | 17.22 | 35.84 | 14.55 | 27.72 | 12.11 | 21.91 | 11.25 | 45.85 | 23.39 | 37.55 | 22.12 |
| BERT4Rec | 50.12 | 17.16 | 36.78 | 15.61 | 28.12 | 12.85 | 22.38 | 11.58 | 46.72 | 25.52 | 38.92 | 23.44 |
| SR-GNN | 50.73 | 17.59 | 36.86 | 15.52 | 27.57 | 13.72 | 23.41 | 13.45 | 50.32 | 26.57 | 43.21 | 26.07 |
| NISER | 53.39 | 18.72 | 40.20 | 17.82 | 33.79 | 16.67 | 28.46 | 16.38 | 54.90 | 29.89 | 47.69 | 29.38 |
| LESSR | 51.71 | 18.15 | 36.16 | 15.64 | 27.88 | 12.08 | 22.68 | 11.68 | 53.05 | 28.01 | 45.76 | 27.51 |
| GCE-GNN | 54.22 | 19.04 | 41.16 | 18.15 | 33.42 | 15.24 | 28.01 | 15.08 | 50.60 | 25.39 | 43.53 | 24.89 |
| $S^2$-DHCN | 53.66 | 18.51 | 40.21 | 17.59 | 31.42 | 15.05 | 26.22 | 14.60 | 53.66 | 27.30 | 46.15 | 26.85 |
| COTREC | 54.18 | 19.07 | 41.88 | 18.16 | 36.35 | 18.04 | 30.62 | 17.65 | 56.17 | 29.97 | 48.61 | 29.46 |
| Int-GNN | 55.16 | 19.46 | 41.84 | 18.53 | 40.77 | 18.20 | 34.28 | 17.74 | 58.02 | 31.48 | 50.41 | 30.94 |
| GCACL-Rec | **55.48** | **19.81** | **42.33** | **18.89** | **42.30** | **19.47** | **35.79** | **19.02** | **58.98** | **32.52** | **51.19** | **32.00** |
| Imp | 0.65% | 1.91% | 1.31% | 2.05% | 4.16% | 7.69% | 4.77% | 7.09% | 1.79% | 3.67% | 1.71% | 3.80% |

**Note**: Results for SASRec and BERT4Rec are sourced from [54] to ensure a fair comparison under consistent data preprocessing and splitting standards. The best-performing results are labeled in bold, and the relative performance improvement of our model compared to the best-performing baseline is labeled with 'Imp'.

To overcome the limitations of traditional methods, recent research has shifted toward graph neural network (GNN) based modeling. SR-GNN employs gated graph neural networks to capture pairwise item transitions within sessions, but it often recommends semantically related yet incorrect items (e.g., predicting a charger instead of a phone case) due to the lack of higher-order dependency modeling. NISER normalizes item and session embeddings to reduce popularity bias, but under sparse sessions, it struggles to retain fine-grained sequential cues, leading to context-mismatched predictions. GCE-GNN extends graph modeling across sessions through session-level and global graphs, but it tends to overemphasize global co-occurrence while neglecting intra-session transitions. $S^2$-DHCN leverages hypergraphs for high-order relations, yet without explicit transition direction, it fails to recover next-item flow, and under sparsity, high-order co-occurrence amplifies noise. COTREC enhances robustness with dual graph views and contrastive co-training, but struggles with nonlinear or complex user intents beyond its consistency objective. Int-GNN integrates item frequency and re-interaction intervals for intent inference, but becomes unstable in short or anonymous sessions where global statistics are insufficient.

Compared with the above baseline methods, the proposed GCACL-Rec demonstrates consistent advantages across all datasets and metrics. On average, GCACL-Rec outperforms the best baseline by 1.48% on Diginetica, 5.92% on Tmall, and 2.74% on RetailRocket. This gain mainly comes from two components: (i) a global, multi-scale graph with hypernodes that captures high-order item transitions within and across sessions, together with a relative multi-head attention mechanism that enhances cross-session propagation by modeling position-sensitive dependencies, thereby alleviating errors such as "associated but non-target" predictions; and (ii) multi-view adversarial contrastive learning coupled with a neural decision forest, which improves robustness under sparsity, mitigates popularity bias, and models non-linear intent patterns more effectively. The performance differences across datasets can be attributed to their distinct characteristics (Table 1). Diginetica has shorter sessions, which

reduces the benefit of high-order structural modeling, leading to relatively smaller improvements. By contrast, Tmall and RetailRocket are more sparse and contain larger item vocabularies, where the proposed global multi-scale graph and adversarial contrastive learning are more effective in alleviating sparsity and popularity bias, resulting in larger relative gains. Overall, GCACL-Rec addresses the key limitations observed in prior work—pairwise-only modeling, popularity bias under sparsity, missing transition direction, and instability in short or anonymous sessions—while achieving state-of-the-art results across benchmarks.

## RQ2-Component ablation study

In this section, we analyze the contribution of each component in our model by developing four variant versions: w/o-MSGNN, w/o-RMA, w/o-MPACL, and w/o-NDF. We compare these variants with the original baselines and the full GCACL-Rec model on the Diginetica, Tmall, and RetailRocket datasets. Specifically, in w/o-MSGNN, we remove the multi-scale global graph and the multi-scale GNN module, using only the item view to model session data. In w/o-RMA, we replace the relative multi-head attention used in node-to-node propagation with standard attention. In w/o-MPACL, we remove the multi-view contrastive-adversarial joint learning module. In w/o-NDF, we remove the neural decision forest predictor and use a single softmax layer for computing the final score. The performance of GCACL-Rec and its four ablated variants is reported in Fig 2.

As shown in Fig 2, GCACL-Rec consistently outperforms all four ablated variants and the original baseline across all three datasets, confirming the effectiveness of each component.

Among them, MSGNN is one of the core modules, enabling the propagation of cross-session information and the sharing of global context. Its removal leads to a noticeable drop in performance across all metrics, as it models hierarchical user behavior through multi-scale hypernodes (mean, max, and random sampling) and preserves complex item transition structures, enhancing representation quality. Similarly, replacing the relative multi-head attention (RMA) with standard attention results in performance degradation. RMA dynamically captures temporal dependencies between hypernodes via relative position encoding. Without it, the model struggles to distinguish between local and global dependencies, losing the ability to adaptively weight hypernode interactions. Removing the MPACL module also results in a noticeable decrease in performance. By combining multi-view contrastive learning and adversarial perturbation, MPACL helps capture diverse user interests and improves robustness against complex data distributions. Excluding the NDF module weakens the model's ability to handle nonlinear patterns. NDF enhances high-order behavior modeling through an ensemble of neural decision trees, supported by James-Stein smoothing and dynamic pruning, which together balance expressiveness and generalization.

In summary, GCACL-Rec integrates hierarchical temporal modeling (MSGNN-RMA), multi-view enhancement (MPACL), and decision optimization (NDF) into a unified "representation–fusion–decision" framework. These components complement each other and work jointly during training, leading to significant improvements in session-based recommendation performance.

## RQ3-Effectiveness study of MSGNN

In this section, we validate the overall effectiveness of our proposed global graph structure and multi-scale graph neural network (MSGNN) by constructing three structural variants: w/o-GCN, w/o-GAT, and w/o-GGNN. Specifically, in these variants, we modify the global graph computation method by replacing our original MSGNN module with classical GNN architectures—namely graph convolutional networks (GCN), graph attention networks

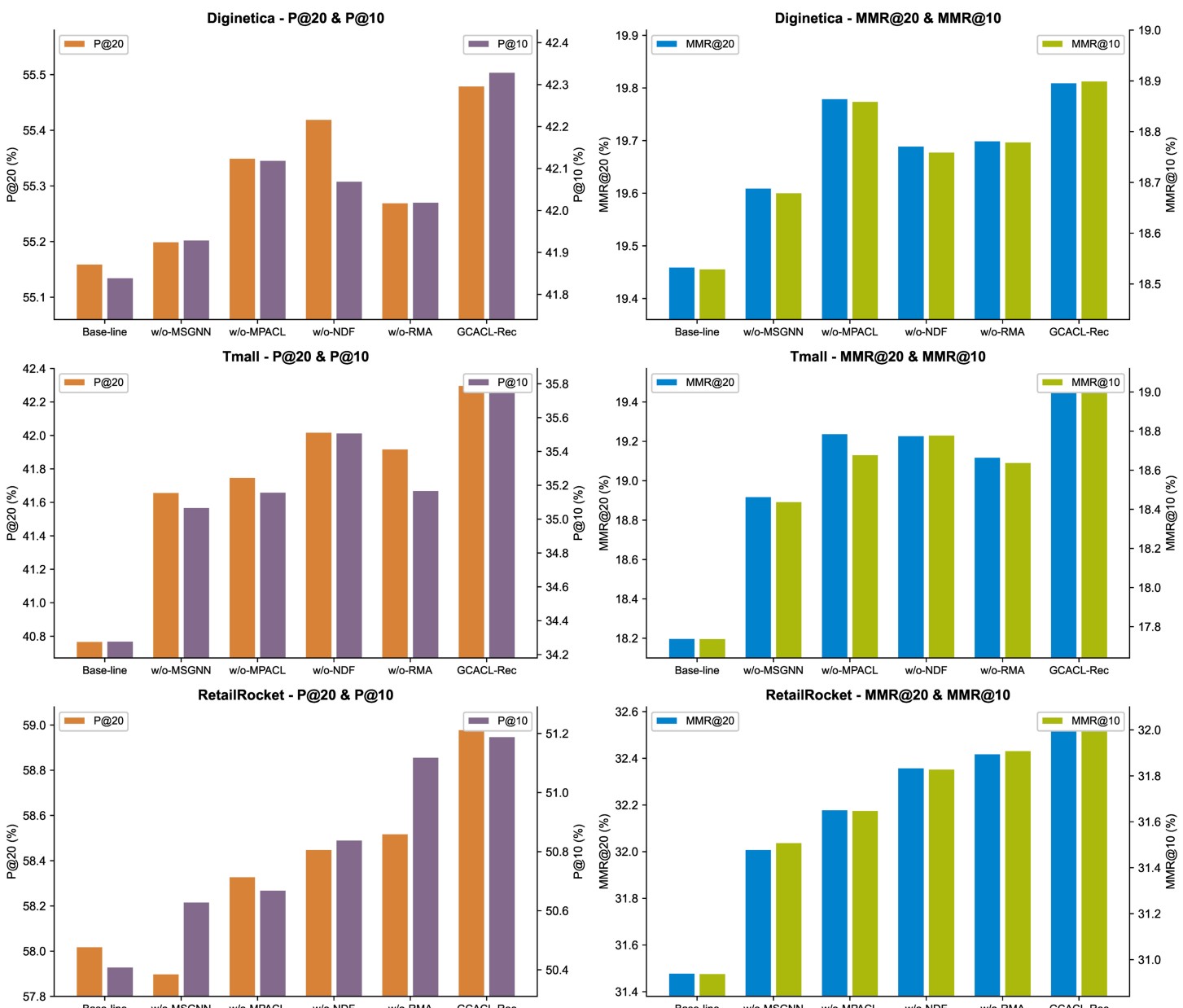

**Fig 2. Changes in GCACL-Rec model performance after ablation of individual components.** The left side of the figure displays the results of the baseline model, while the right side shows the results of our model. The four modules in the middle represent the results of ablating each component.

(GAT), and gated graph neural networks (GGNN)—to encode the item graph and perform information propagation. All other components of the model remain unchanged. This allows us to evaluate the adaptability and performance differences of various GNNs in the context of graph modeling.

As shown in Fig 3, the GCN, GAT, and GGNN variants can capture some cross-session structural information and demonstrate a certain level of effectiveness. However, compared to our proposed global module, a performance gap remains across key metrics on all three datasets. For example, on the Tmall dataset, MSGNN outperforms GAT and GGNN by

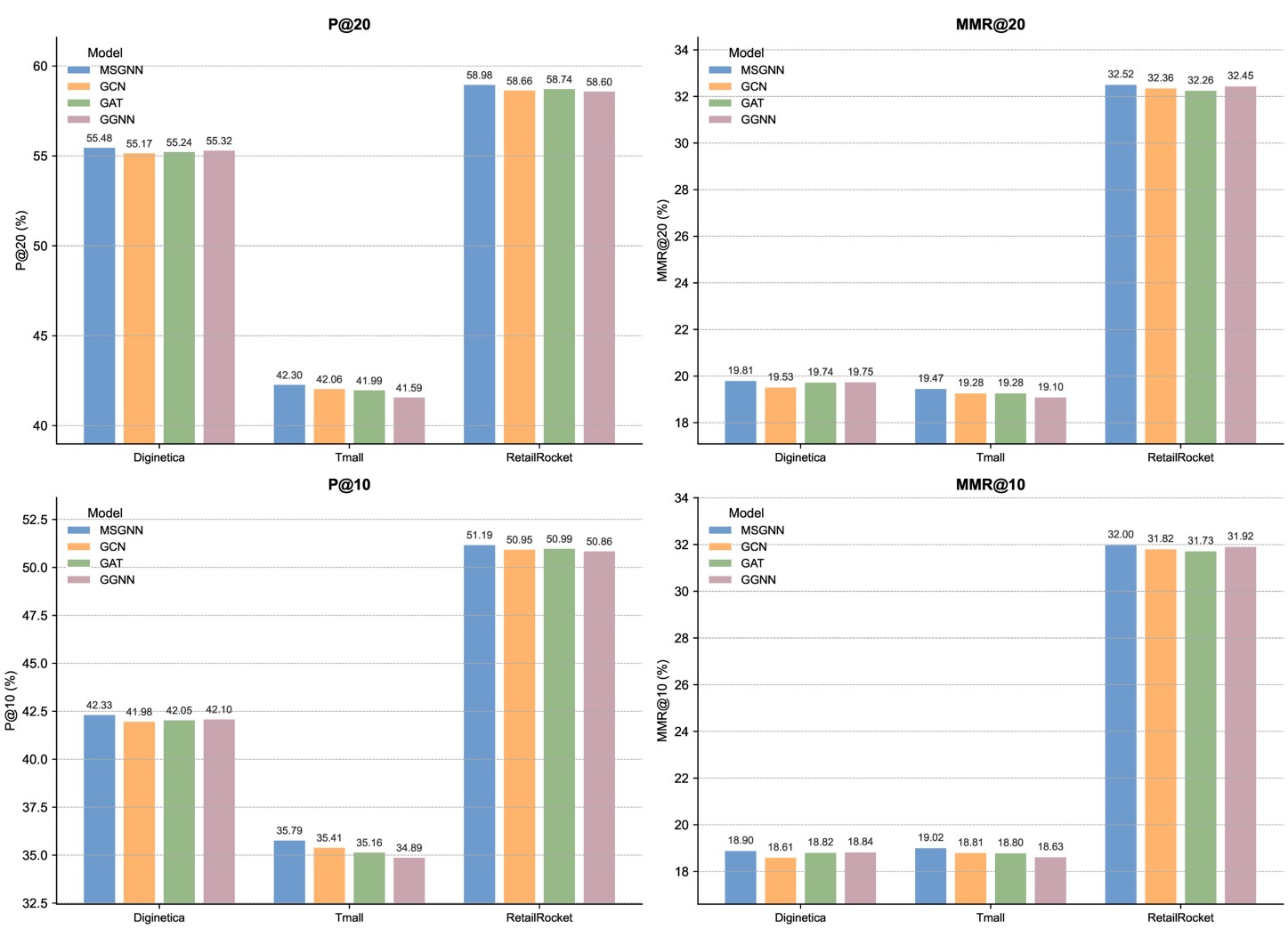

**Fig 3. Performance of different GNN variants.** The figure illustrates the effectiveness of our multi-scale global graph design by comparing it with classical graph neural network (GNN) backbones.

approximately 0.6% and 0.9% on P@20, respectively, and achieves improvements of 0.8%–1.1% on MMR@20. Similar trends are observed on the RetailRocket and Diginetica datasets, with especially notable gains in MMR. The key reason for this performance difference lies in the design of our global graph, which not only captures standard item transition structures but also introduces hypernodes and multi-scale aggregation to model higher-order, hierarchical relationships across sessions. Moreover, MSGNN enables multi-round bidirectional message passing between nodes, hypernodes, and among hypernodes themselves, significantly enhancing the expressive power of the global graph. In contrast, the alternative models lack this structural support; their message propagation is limited to shallow or static neighborhood connections, making it difficult to capture complex behavioral paths and latent semantic associations effectively.

In summary, while traditional GNNs such as GCN, GAT, and GGNN demonstrate some modeling capacity in our cross-session global information modeling task, MSGNN—designed

specifically for the proposed global graph structure—offers a better fit for multi-scale representation and consistently delivers more stable and superior recommendation performance.

## RQ4-Effect of the number of super points

To investigate the impact of the number of hypernodes on the performance of the MSGNN model in the session-based recommendation, we conducted experiments on the Diginetica, RetailRocket, and Tmall datasets, varying the number of hypernodes in the set 2, 4, 6, 8, 10, 12. Fig 4 presents the results, with the left side showing changes in P@20 and P@10 and the right side showing trends in MMR@20 and MMR@10.

When the number of hypernodes is set to 2, the model captures only coarse-grained semantics, such as global averages and local peaks. This limits its ability to aggregate useful information, resulting in lower performance across all metrics. With four hypernodes, the model achieves the best results on all datasets in both P@K and MMR@K, suggesting an effective balance between generalization and recommendation quality. This setting enhances information propagation without causing excessive aggregation. However, increasing the

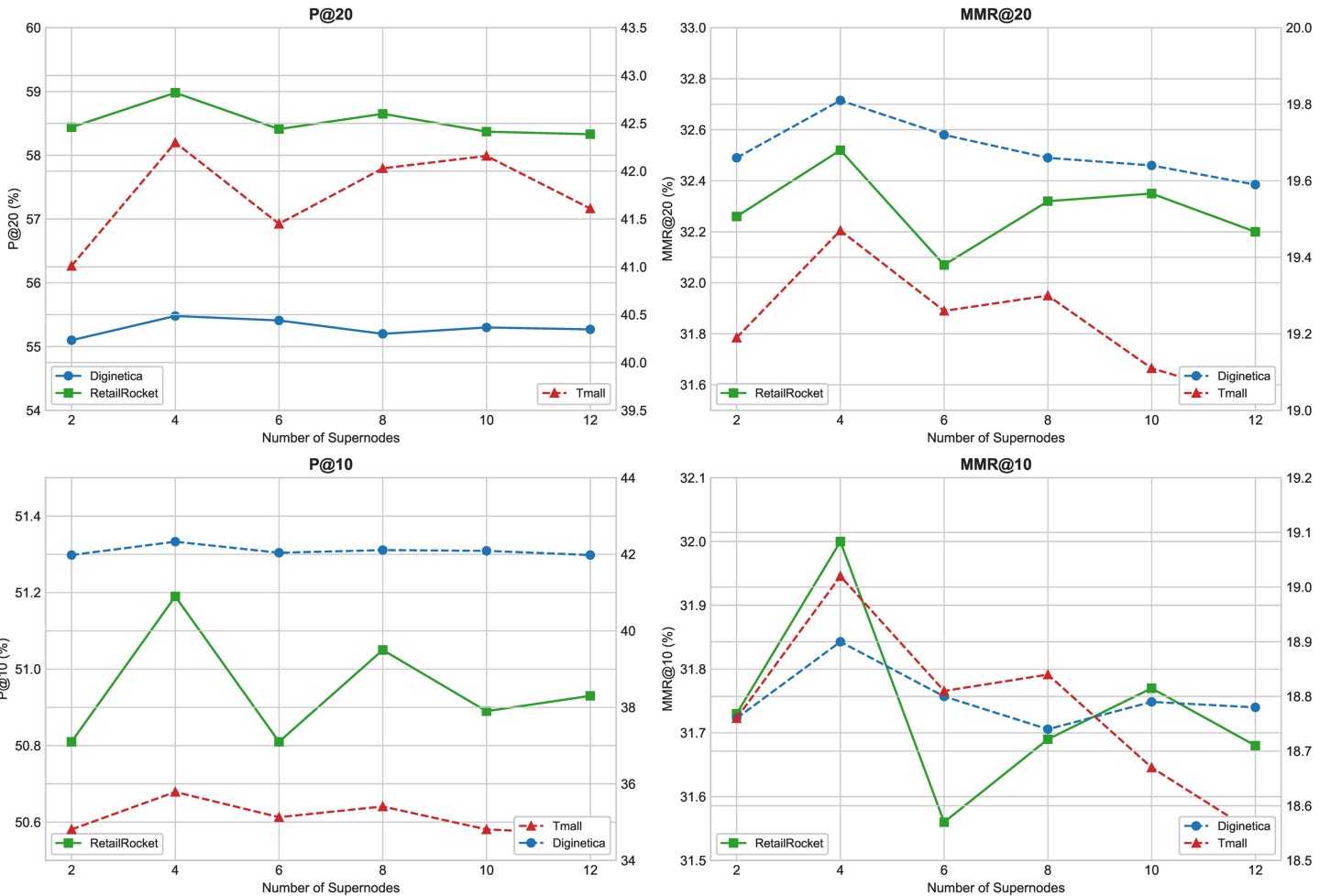

**Fig 4. Impact of the number of hypernodes on model performance on three datasets.** The figure shows how varying the number of hypernodes impacts accuracy (P@20/P@10) and diversity (MMR@20/MMR@10), demonstrating that using four hypernodes yields the best overall performance.

number of hypernodes beyond 4 (from 6 to 12) leads to performance degradation or instability. This may be due to overlapping semantics among hypernodes after multiple rounds of message passing, which introduces interference and reduces model expressiveness.

In conclusion, the number of hypernodes has a significant impact on the performance of MSGNN. Too few may lead to the under-representation of structural information, while too many can cause redundancy and noise. Setting the number of hypernodes to 4 offers the best trade-off for the effective session-based recommendation.

### RQ5-Impact of model depth

To systematically evaluate the representational capacity of the model under different propagation depths, we conducted a series of joint ablation experiments by configuring both the local graph network (P-GNN) and the global graph network (Global-GNN) with 1 to 5 layers. These experiments were performed on the Diginetica, Tmall, and RetailRocket datasets, and the results are illustrated in Fig 5. In each experiment group, the local and global networks were kept at the same depth to ensure comparability.

Overall, the model exhibits relatively weak performance at shallow depths (e.g., one layer). As the number of layers increases, both recommendation accuracy (P@20, P@10) and mean

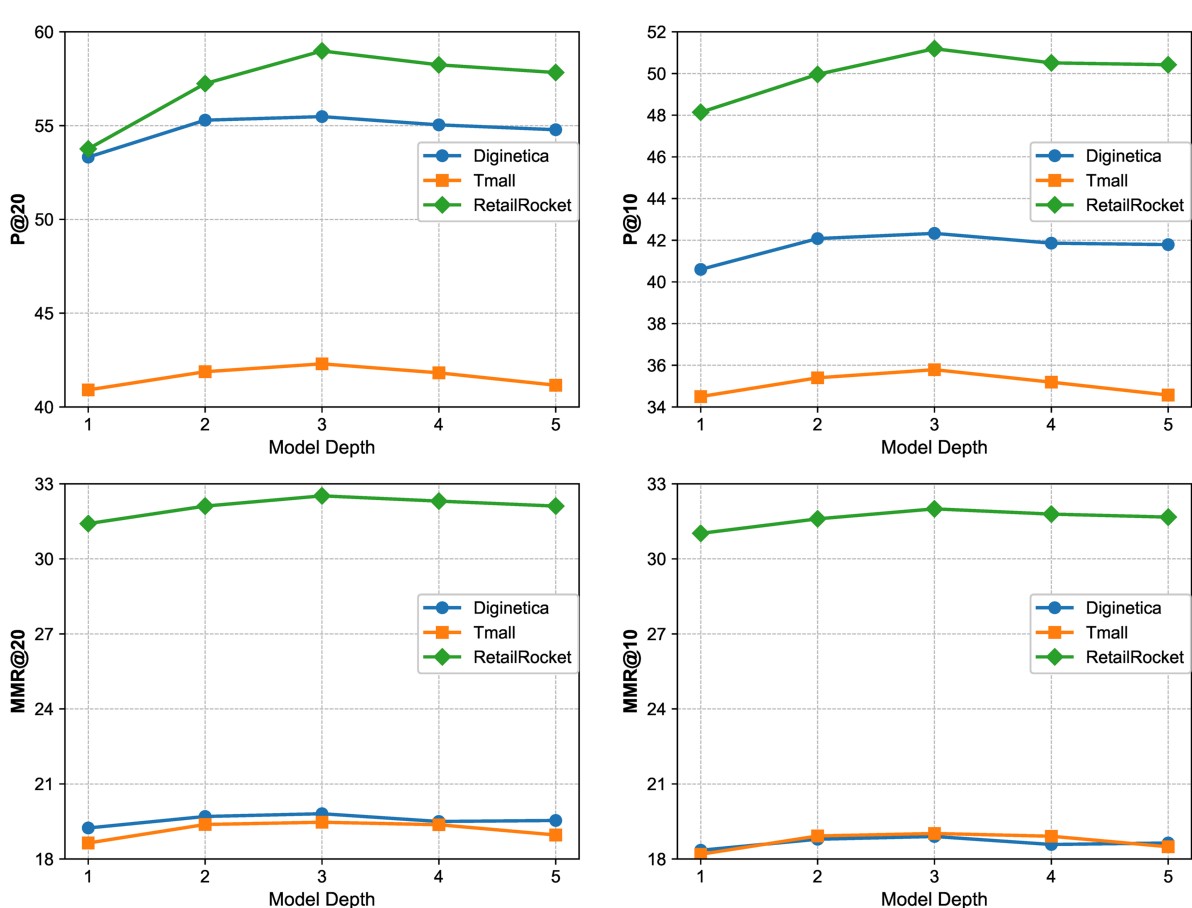

**Fig 5. Impact of varying propagation depth on model performance.** The figure compares the performance of GCACL-Rec with different numbers of layers in both local and global GNNs, showing that a three-layer setting yields the best accuracy.

reciprocal rank (MMR@20, MMR@10) improve steadily, reaching their peak at three layers. Specifically, on the Diginetica dataset, the three-layer model achieves the best results, with P@20 reaching 55.48 and MMR@20 reaching 19.81. Similarly, the three-layer configuration yields optimal results across the main metrics on both Tmall and RetailRocket datasets. Notably, when the model depth exceeds three layers, we observe slight fluctuations or even minor performance drops. This degradation can be attributed to the over-smoothing effect or noise accumulation introduced by excessively deep networks, which leads to homogenized node representations and diminished structural discriminability. Furthermore, deeper propagation may introduce redundant dependencies or cause gradient vanishing, thereby weakening the model's ability to capture both local and global semantics effectively.

In summary, a three-layer propagation structure strikes a favorable balance between accuracy and diversity. It is sufficient to capture higher-order structural information while effectively preventing overfitting and representational degradation, thus delivering the best overall performance. Therefore, we adopt a three-layer architecture as the default setting in the final model to strike a balance between expressive power and computational efficiency.

## RQ6-Sensitivity analysis

To validate the rationality of our chosen hyperparameters, we first conducted a sensitivity experiment on the learning rate. On the RetailRocket dataset, we tested four values of learning rate (lr) at 0.001, 0.0015, 0.002, and 0.025, and plotted the curves of Recall@20 and MRR@20 over epochs (in Fig 6). The results show that a smaller learning rate (0.001) ensures stable convergence but converges more slowly overall; moderate learning rates (0.0015 and 0.002) lead to rapid performance gains in the early epochs and eventually achieve higher Recall and MRR, with lr = 0.0015 performing best, reaching about 58.98 in Recall@20 and 32.52 in MRR@20. In contrast, a substantial learning rate (0.025) improves quickly in the early stage but exhibits significant fluctuations later, indicating instability in training. Thus, lr = 0.0015 strikes a good balance among convergence speed, final performance, and stability, making it the preferable choice.

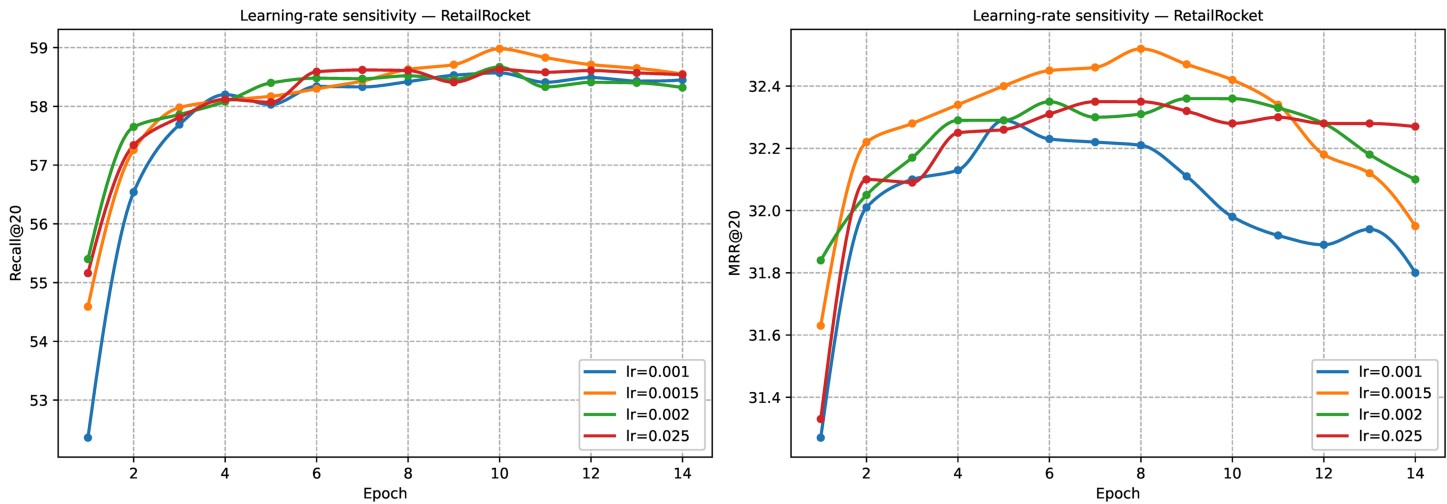

**Fig 6. Impact of learning rate on model performance.** The figure shows the results on RetailRocket with different learning rates, where lr = 0.0015 yields the best accuracy.

We further performed sensitivity analysis on the contrastive loss parameter $\mu$, with results shown in Table 3. When $\mu = 0.10$ and $\mu = 0.15$, the model achieved superior and stable performance, with $\mu = 0.10$ performing best (Recall@20 = 55.48, MRR@20 = 19.81). In contrast, when $\mu = 0.20$ and $\mu = 0.25$, performance dropped noticeably, with Recall@20 falling to around 53.74. This suggests that an overly large $\mu$ assigns excessive weight to the contrastive loss, thereby weakening the optimization of the primary recommendation task. Considering both performance and stability, $\mu = 0.10$ is the optimal setting. Overall, the final parameter configuration (lr = 0.0015, $\mu = 0.10$) was determined through extensive empirical validation and ensures consistent and strong performance on both Recall and MRR, thus supporting the soundness of our parameter choices.

## Conclusion

This study proposes GCACL-Rec, a session-based recommendation model that combines global graph modeling, multi-scale graph neural networks, and multi-perspective contrastive adversarial learning. GCACL-Rec constructs a multi-scale global graph with super-nodes to capture cross-session semantics and uses relative multi-head attention to enhance directional message passing. For local sessions, position encoding and gating mechanisms capture short-term dependencies. The MPACL framework further strengthens user intent representation. Experiments on Diginetica, Tmall, and RetailRocket show that GCACL-Rec consistently outperforms existing methods in accuracy and diversity, while maintaining stability under sparsity and cold-start conditions.

Despite its strong performance, GCACL-Rec has some limitations. First, the integration of global graphs, multi-scale attention, and adversarial training increases computational complexity, which may reduce training efficiency on large-scale datasets. Second, the construction of super-nodes and message passing depends on manually designed structures and aggregation strategies, which limit adaptability and generalization in highly dynamic scenarios.

For future work, we plan to mitigate computational complexity by applying model compression techniques such as pruning and quantization, reducing model size, and improving inference speed without sacrificing performance. We will also explore distributed training with multi-GPU setups and cloud platforms to accelerate training and scale to larger datasets. These efforts are essential for enhancing the efficiency and real-world applicability of the proposed model. In summary, GCACL-Rec provides an effective solution for cross-session fusion, high-order behavior modeling, and self-supervised learning, offering a strong foundation for personalized recommendations in complex scenarios. If access to the code is required, interested readers may contact the corresponding author, and access may be granted upon reasonable request and evaluation.

**Table 3**. **Performance comparison under different $\mu$ values on Diginetica.**

| $\mu$ | Recall@20 | MRR@20 | Recall@10 | MRR@10 | Best Epoch |
|---|---|---|---|---|---|
| 0.10 | **55.45** | **19.84** | **42.15** | **18.92** | 11 |
| 0.15 | 55.33 | 19.81 | 42.08 | 18.89 | 10 |
| 0.20 | 54.87 | 19.68 | 41.90 | 18.78 | 7 |
| 0.25 | 53.74 | 19.29 | 41.03 | 18.40 | 3 |

**Note**: This table shows the performance of GCACL-Rec on the Diginetica dataset under different values of $\mu$. The best result is obtained when $\mu = 0.10$.

## Acknowledgments

I want to express my sincere gratitude to all those who have supported and assisted me throughout the course of this research. First, I am deeply grateful to my advisor for the valuable guidance, insightful suggestions, and continuous encouragement throughout the research development and manuscript preparation. Your rigorous academic attitude and profound expertise have greatly inspired me. I also extend my appreciation to my lab colleagues for their assistance and constructive feedback, which contributed significantly to the successful implementation and testing of the model. I am especially grateful to my family for their unwavering understanding and support, which has been a constant source of motivation. Finally, I would like to thank the providers of the datasets and the authors of the referenced works, whose contributions laid the groundwork for this research.

## Author contributions

**Conceptualization:** Xiaowen Liu, Xinhuan Chen, Ming Ma.

**Formal analysis:** Xiaowen Liu.

**Funding acquisition:** Ming Ma.

**Investigation:** Xiaowen Liu, Ming Ma.

**Methodology:** Xianghui Li, Xinhuan Chen.

**Project administration:** Xianghui Li.

**Resources:** Xianghui Li.

**Software:** Xianghui Li, Xiaowen Liu.

**Supervision:** Xianghui Li, Xinhuan Chen, Ming Ma.

**Validation:** Ming Ma.

**Visualization:** Xianghui Li.

**Writing – original draft:** Xianghui Li.

**Writing – review & editing:** Xianghui Li.

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
