## [Decision Letter · Decision Letter 0]

28 Jul 2025

PONE-D-25-30866MSGNN: A Study on Conversational Recommendation via Global Context-Aware and Multi-View Contrastive Adversarial Joint LearningPLOS ONE

Dear Dr. Ma,

Thank you for submitting your manuscript to PLOS ONE. After careful consideration, we feel that it has merit but does not fully meet PLOS ONE’s publication criteria as it currently stands. Therefore, we invite you to submit a revised version of the manuscript that addresses the points raised during the review process.

We look forward to receiving your revised manuscript.

Kind regards,

Ping Xiang

Academic Editor

PLOS ONE

Journal Requirements:

4. Thank you for stating in your Funding Statement: [This work was supported in part by the National Natural Science Foundation of China under Grant 42004153, and in part by the 2024 Graduate Student Innovation Program of Beihua University under Grant 2024058.].

Reviewers' comments:

Reviewer's Responses to Questions

**Comments to the Author**

1. Is the manuscript technically sound, and do the data support the conclusions?

Reviewer #1: Yes

Reviewer #2: Yes

2. Has the statistical analysis been performed appropriately and rigorously? 

Reviewer #1: Yes

Reviewer #2: Yes

3. Have the authors made all data underlying the findings in their manuscript fully available?

Reviewer #1: No

Reviewer #2: Yes

4. Is the manuscript presented in an intelligible fashion and written in standard English?

Reviewer #1: Yes

Reviewer #2: Yes

5. Review Comments to the Author

Reviewer #1: Review comments:

The manuscript presents a novel session-based recommendation framework integrating global context-aware graph modeling with multi-view contrastive adversarial learning. The overall technical design is comprehensive and demonstrates clear performance gains over competitive baselines. However, several key aspects require further refinement, including clarification of model motivation, comparative analysis with recent transformer-based methods, deeper theoretical justification for design choices (e.g., MSGNN and NDF), and improvement in figure quality. Moreover, the literature review would benefit from incorporating recent relevant works on graph-based attention and predictive modeling in complex sequential systems.

In its current form, the paper demonstrates potential, but the above-mentioned concerns should be thoroughly addressed to enhance the scientific clarity, methodological rigor, and academic completeness. I recommend a major revision before the manuscript can be considered for publication.

Comment 1:

The proposed GCACL-Rec model is ambitious and multi-component, but the overall model description would benefit from a clearer visual pipeline or algorithm box that concisely summarizes each stage (MSGNN, MPACL, NDF). This would help readers track the execution flow.

Comment 2:

Terms like “supernodes”, “hypernodes”, and “global graph” are used interchangeably. Standardizing this terminology and releasing training scripts or code snippets would significantly improve reproducibility.

Comment 3:

Several figures in the manuscript (e.g., Fig. 1: overall model architecture, Fig. 2–5: ablation and comparison results) appear blurry and lack sufficient resolution when viewed in print or on screen. This significantly affects the readability of key visual elements such as graph legends, axis labels, and numerical values. The authors are strongly encouraged to regenerate all figures with higher resolution (preferably vector graphics such as PDF, SVG, or EPS formats) to ensure visual clarity and professional presentation quality.

Comment 4:

Missing Relevant Graph-Based Literature

The manuscript could be strengthened by referencing recent graph-based models applied to dynamic system prediction and safety assessment. In particular, the following works provide relevant insights into graph modeling and attention mechanisms in sequential or infrastructure systems and would help contextualize the contribution of MSGNN in the broader research landscape:

Graph-based attention model for predictive analysis in train-bridge systems, Applied Soft Computing, 2025, https://doi.org/10.1016/j.asoc.2025.113360

Enhanced multi-scenario running safety assessment of railway bridges based on graph neural networks with self-evolutionary capability, Engineering Structures, 2024, https://doi.org/10.1016/j.engstruct.2024.118785

These papers offer valuable perspectives on graph-based attention mechanisms and safety prediction in complex, sequential environments. The authors are encouraged to cite and briefly discuss them to enhance the relevance and completeness of the related work section.

Reviewer #2: The paper is innovative in proposing the GCACL-Rec model in the study of session recommender systems, integrating multi-scale graph neural networks and comparative learning frameworks, but its research content needs to be further improved, for example:

1. The abstract states that a new model of GCACL-Rec is proposed, but the title, body and conclusion refer to “MSGNN” several times (e.g. “MSGNN: A Study on...”)., please explain briefly. In addition, please explain whether the first appearance of GCACL-Rec is defined and how the most central innovation of the model is reflected.

2. Are hyperparameter settings (e.g., number of super-nodes, model depth) determined through systematic tuning experiments? Are parameter sensitivity analyses (e.g., learning rate, number of layers vs. performance curve) provided to support the rationality of the current choices?

3. Is the “increase in computational complexity” mentioned in the model limitations section quantified by a time complexity analysis? Are potential optimization directions discussed (e.g., model compression, distributed training)?

4. The manuscript reports overall performance but does not analyze error cases in depth. Can you provide a qualitative analysis of high-frequency error patterns (e.g., mistakenly pushing associated items)?

5. Is the “stable performance in sparse and cold start scenarios” mentioned in the conclusion based on sufficient experimental evidence? Are comparative performance data for specific scenarios provided?

6. Table 2 shows that its performance metrics behave erratically on different datasets. Please explain its rationale.

7. Does the baseline model selection cover different technical routes (e.g., RNN, GNN, Transformer-based approaches)? In addition, it is necessary to cite session recommendations of recent reviews or GNN comparative learning frontier work to highlight the work of this study.

6. PLOS authors have the option to publish the peer review history of their article (what does this mean?). If published, this will include your full peer review and any attached files.

Reviewer #1: No

Reviewer #2: No

---

## [Decision Letter · Decision Letter 1]

7 Oct 2025

GCACL-Rec: A Study on Conversational Recommendation via Global Context-Aware and Multi-View Contrastive Adversarial Joint Learning

PONE-D-25-30866R1

Dear Dr. Ma,

We’re pleased to inform you that your manuscript has been judged scientifically suitable for publication and will be formally accepted for publication once it meets all outstanding technical requirements.

Kind regards,

Ping Xiang

Academic Editor

PLOS ONE

Additional Editor Comments (optional):

Reviewers' comments:

Reviewer's Responses to Questions

**Comments to the Author**

1. If the authors have adequately addressed your comments raised in a previous round of review and you feel that this manuscript is now acceptable for publication, you may indicate that here to bypass the “Comments to the Author” section, enter your conflict of interest statement in the “Confidential to Editor” section, and submit your "Accept" recommendation.

Reviewer #1: (No Response)

Reviewer #3: (No Response)

2. Is the manuscript technically sound, and do the data support the conclusions?

Reviewer #1: (No Response)

Reviewer #3: (No Response)

3. Has the statistical analysis been performed appropriately and rigorously? 

Reviewer #1: (No Response)

Reviewer #3: (No Response)

4. Have the authors made all data underlying the findings in their manuscript fully available?

Reviewer #1: (No Response)

Reviewer #3: (No Response)

5. Is the manuscript presented in an intelligible fashion and written in standard English?

Reviewer #1: (No Response)

Reviewer #3: (No Response)

6. Review Comments to the Author

Reviewer #1: (No Response)

Reviewer #3: (No Response)

7. PLOS authors have the option to publish the peer review history of their article (what does this mean?). If published, this will include your full peer review and any attached files.

Reviewer #1: No

Reviewer #3: No

---

## [Editor Report · Acceptance letter]

PONE-D-25-30866R1

PLOS ONE

Dear Dr. Ma,

I'm pleased to inform you that your manuscript has been deemed suitable for publication in PLOS ONE. Congratulations! Your manuscript is now being handed over to our production team.

Kind regards,

on behalf of

Professor Ping Xiang

Academic Editor

PLOS ONE